# Artificial Neural Networks (ANNs) for Vapour-Liquid-Liquid Equilibrium (VLLE) Predictions in N-Octane/Water Blends

Esteban Lopez-Ramirez [1,2], Sandra Lopez-Zamora [1], Salvador Escobedo [1] and Hugo de Lasa [1,*]

1   Department of Chemical and Biochemical Engineering, Chemical Reactor Engineering Centre,
    The University of Western Ontario, London, ON N6A 3K7, Canada
2   Faculty of Engineering and Architecture, Department of Civil Engineering,
    Universidad Nacional de Colombia, Manizales 170004, Colombia
*   Correspondence: hdelasa@uwo.ca; Tel.: +1-519-661-2144

**Abstract:** Blends of bitumen, clay, and quartz in water are obtained from the surface mining of the Athabasca Oil Sands. To facilitate its transportation through pipelines, this mixture is usually diluted with locally produced naphtha. As a result of this, naphtha has to be recovered later, in a naphtha recovery unit (NRU). The NRU process is a complex one and requires the knowledge of Vapour-Liquid-Liquid Equilibrium (VLLE) thermodynamics. The present study uses experimental data, obtained in a CREC-VL-Cell, and Artificial Intelligence (AI) for vapour-liquid-liquid equilibrium (VLLE) calculations. The proposed Artificial Neural Networks (ANNs) do not require prior knowledge of the number of vapour-liquid phases. These ANNs involve hyperparameters that are used to obtain the best ANN model architecture. To accomplish this, this study considers (a) $R^2$ Coefficients of Determination and (b) ANN training requirements to avoid data underfitting and overfitting. Results demonstrate that temperature has a major influence on ANN vapour pressure predictions, while the concentration of octane, the naphtha surrogate having, in contrast, a lesser effect. Furthermore, the ANN data obtained allows the calculation of octane-in-water and water-in-octane maximum solubilities.

**Keywords:** hydrocarbon/water blends; Artificial Neural Networks; vapour-liquid-liquid equilibrium; Machine Learning

## 1. Introduction

Canada is the sixth largest oil producer in the world, significantly contributing to the Athabasca oil, tar, and bituminous sands [1]. Northern Alberta holds one of the world's largest deposits of hydrocarbons, containing more than 175 billion barrels of bitumen [2]. Bitumen from the oil sands is produced by employing approximately 20% surface mining and 80% in situ technologies. Surface mining yields a blend of bitumen, quartz, clay, and water. This is composed of about 85% quartz/clay particles and 15% bitumen plus water [3]. There are three classes of oil sands with different bituminous contents: low-grade (6–8 wt% bitumen), medium-grade (8–10 wt% bitumen), and rich grade (>10 wt% bitumen) oil sands [3]. Bitumen is composed of a complex mixture of hydrocarbons, with its elemental composition including hydrogen, carbon, nitrogen, and metals such as vanadium and nickel [3]. Bitumen can be blended with a solvent to remove water and solids. Usually, this solvent is a locally produced naphtha [4]. Following this process, the naphtha can be reclaimed in a Naphtha Recovery Unit (NRU). In this manner, one is able to reuse it, while minimizing its environmental impact [4,5].

In the NRU feed, hydrocarbons are blended with water. Due to the low hydrocarbon miscibility in water, and the lack of Vapour-Liquid-Liquid Equilibrium (VLLE) data available, our research group has shown that there is presently a lack of predictability of the NRU recovery unit efficiency [5–8]. Thus, studies considering hydrocarbon/water blends, and their vapour-liquid-liquid equilibrium (VLLE), as shown in Figure 1, are of major significance to improve the design and the operation of these VLL separators.

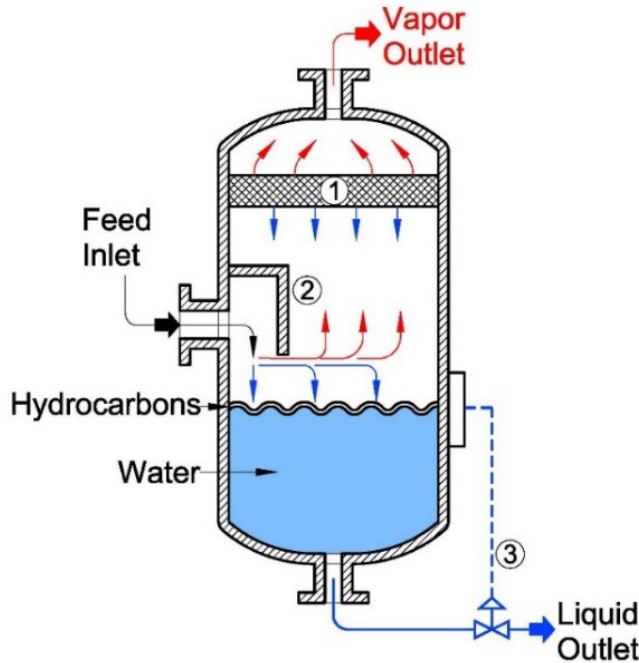

① De-entrainment Mesh Pad
② Inlet Diverter
③ Liquid Level Control Valve

**Figure 1.** Schematic description of a typical VLL Separator.

With the advancement of computational power, ANNs have attracted the attention of the scientific community, in order to solve VLLE and flash calculation problems. Schmitz et al. (2006) used ANNs to predict the number of phases in a ternary ethanol–ethyl acetate–water blend [6]. A total of 58,828 experimental data points were used concurrently with the NRTL model and the Antoine equations. This was conducted to evaluate liquid–liquid equilibrium, vapor–liquid–liquid equilibrium, and vapor–liquid equilibrium, at temperatures ranging from 335.2 K to 373 K, at 760 mmHg [9]. Argatov and Kocherbitov (2019) proposed the generalization of Wilson and NRTL models by employing ANNs and acetic acid–n-propyl alcohol–water blends at 313.15 K [10]. Li et al. applied ANNs to binary blend flash calculations by using a set of 1300 data points [11]. Zhang et al. (2020) used ANNs with data generated from an iterative NVT flash calculation. They suggested that by model training reliability was questionable. Thus, as an alternative, they proposed a self-adaptive algorithm to accelerate the flash calculations. The critical properties of each component were considered as inputs to the neural network, with the output identifying the total number of phases at equilibrium and the molar compositions in each phase [12]. For ANN model training and testing, 90,601 data points obtained from an 8-hydrocarbon blend were employed. A total of 90% of the data were used for training and a 10% was employed for ANN model validation [3].

In previous work, our research group [5] employed Machine Learning (ML) methods to calculate VLE and VLLE by using theoretical data from traditional thermodynamic models such as the NRTL [6,8]. It was extensively shown that flash and other ML calculations using theoretical data or data from simulations were not reliable [10]. In the present work, ANNs are used instead to calculate VLLE. N-octane in water blends in experiments conducted in a CREC-VL-Cell [5] are employed to train and validate the ANNs. Octane in water is used given that it is a good surrogate to represent naphtha in water blends [9]. The proposed ANNs circumvent convergence problems reported in [6], occurring when using classical VLLE thermodynamics equilibrium models and do not require one to know the number of VL or VLL phases prior to equilibrium calculations [6].

## 2. Materials and Methods

This section describes both octane and water solutions, the CREC-VL-Cell unit used for VLE and VLLE experiments, vapour-liquid and vapour, the data recorded and the Artificial Neural Networks of the present study.

### 2.1. Materials

Distilled water (18.02 g/mol) was used in all the experimental studies. n-Octane (114.23 g/mol) with a 99.0% purity and a 0% water content was obtained from Sigma-Aldrich (St. Louis, MO, USA).

### 2.2. CREC-VL-Cell Unit

The Chemical Reactor Engineering Center (CREC, London, ON, Canada) developed a CREC-VL-Cell unit that allows the VLL equilibrium thermodynamic measurements of hydrocarbon–water blends (refer to Figure 2) [6]. The CREC-VL-Cell uses a marine type of impeller (propeller) that ensures close to isothermal conditions and homogenous mixing inside the cell. This special cell design, proposed by the CREC team, allows one to analyze a hydrocarbon–water sample directly, without losses of light volatile components, due to sample transfers [8].

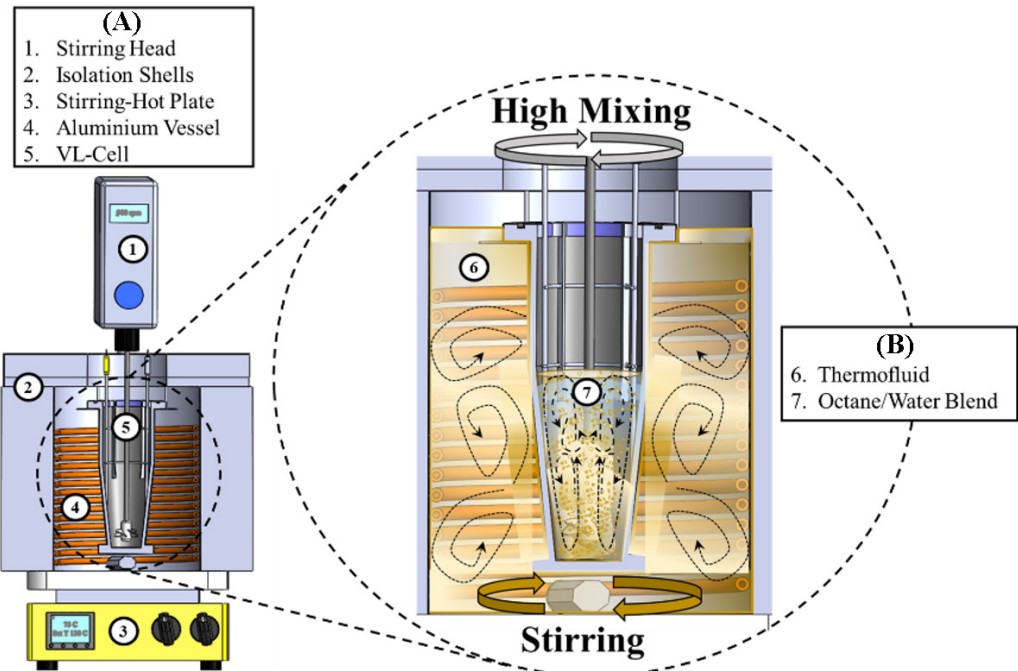

**Figure 2.** Schematic Representation of the CREC-VL-Cell Unit: (**A**) CREC VL Cell Unit, (**B**) Zoom of mixing in the CREC VL Cell. Notes: (1) Stirring Head, (2) Isolation Shells, (3) Stirring Hot Plate, (4) Aluminum Vessel, (5) VL-Cell, (6) Thermofield, and (7) Octane/water Blend [6].

This unit uses a "dynamic heating method" with the temperature of the cell increasing progressively, using a thermal ramp of 1.22 °C/min. As a result, every run provides a large amount of vapour-liquid equilibrium data (10 Hz), with the vapour pressure data being recorded at various temperatures, every 0.01 s.

The proposed dynamic method, as reported in Figure 3, involves the simultaneous recording of temperature and pressure, as the run time is progressing. Additional explanations regarding the cell operation are reported in [5,8]. Data obtained from this dynamic method were validated with static measurements [5,8].

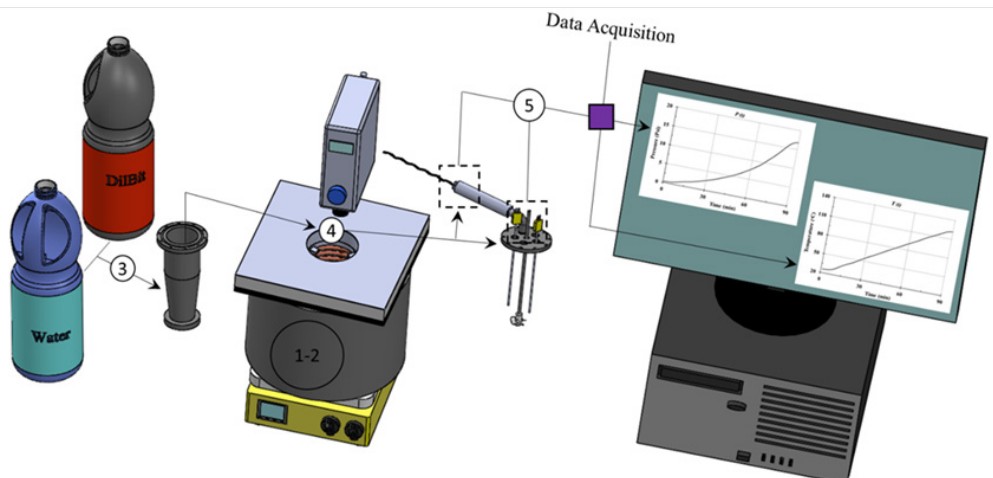

**Figure 3.** Schematic diagram of CREC-VL-Cell Dynamic Experimental Method. Notes: (1,2): stirring hot plate and aluminum vessel, (3) water and octane, (4) CREC-VL Cell, (5) impeller and thermocouples [8].

### 2.3. Data Available and Data Cleansing

Experimental data were obtained in the CREC-VL-Cell of the present study by changing the temperature for a set initial n-octane concentration in water. One should note that for every temperature and initial n-octane in water concentration, the total pressure measured required a correction. To accomplish this, the air partial pressure was discounted from the cell total pressure, as proposed in [5].

Sixty-one (61) runs were developed by using various n-octane/water blends and temperatures. The obtained data were combined into one single file, leading to 4200 run records that were employed to evaluate the proposed ANN models. Additional details regarding the 61 runs developed were reported by our research team in [5].

### 2.4. Artificial Neural Networks (ANN)

ANN models emulate the human brain [13]. The human brain is able to remember and use earlier experiences as a precedent for different future situations. ANNs include basic elements designated as nodes (also called units) that represent brain neurons, as well as layers that represent an ensemble of nodes [14]. Figure 4 describes an ANN configuration with input and output layers. The input and output layers are connected through hidden layers. The input layers provide the data provided to the ANN model, while the hidden layers process the data, and finally, the output layers yield the ANN model output variables [14–16].

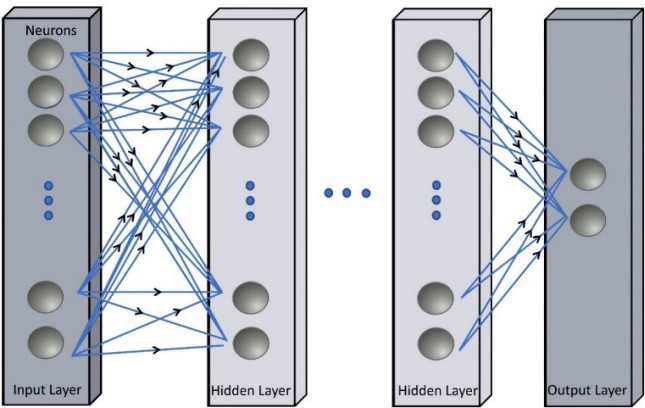

**Figure 4.** Schematic Representation of an Artificial Neural Network Basic Configuration showing Nodes and Layers and their Interactions. Note: The input and output layers are shown in "dark" grey while the hidden layers are displayed in "light" grey.

Figure 5 further describes coexisting inner processes taking place in between "j" and "j + 1" generic hidden layers. One can observe in Figure 5 that both "forward calculations" and "backward calculations" take place. In the "forward calculation", data are fed to the "j" hidden layer, processed in this layer, and then moved from the "j" to the "j + 1" hidden layer. In the "backward calculation", data are fed and processed in the "j + 1" layer and moved back in the opposite direction, from the "j + 1" layer to the "j" hidden layer.

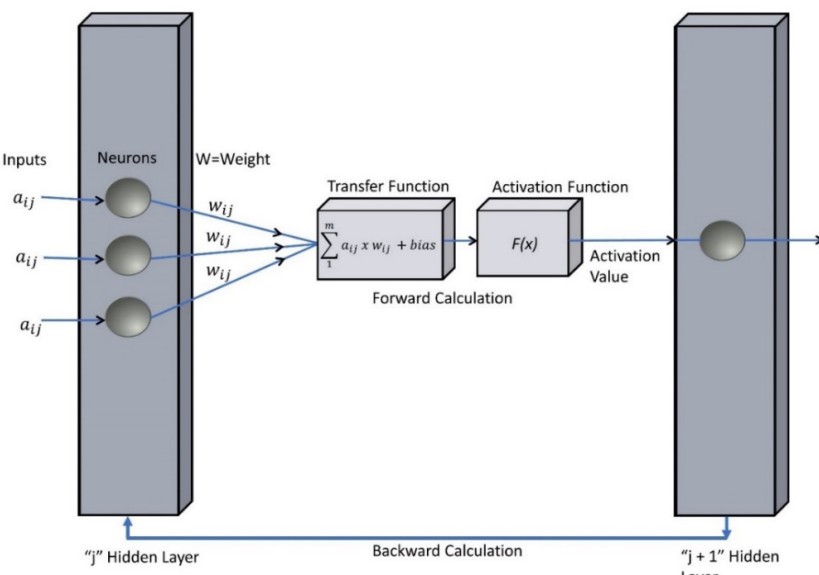

**Figure 5.** Schematic representation of the ANN "Forward and Backward Calculations", taking place in the generic "j" and "j + 1" hidden layers.

Thus, ANN modeling involves Steps A to D that describe the connection between layers in the forward direction, and Steps F to H that explain the ANN connections in the opposite backward direction, as follows:

- *Step A:* A random initial weight is assigned to all nodes in the multilayer ANN model structure [17].
- *Step B:* An "$x$" parameter is defined for each layer connection as $x = \sum_{i=0}^{m} a_{ij} w_{ij} + A$, with "$x$" being the sum of the $a_{ij}$ input, $w_{ij}$ representing the weights assigned in the generic "$j$" hidden layer, "$A$" denoting a bias parameter, and "$m$" standing for the layer number of nodes [18].
- *Step C :* An $\alpha(x) \, F(x)$ activation function is chosen to provide an adequate fitting for the available data [19].
- *Step D:* The revised data calculated in *Step C* is transferred to the next "j + 1" hidden layer. This process is repeated until the very last layer or output layer is reached [20].
- *Step E:* An output error is calculated, using an $R^2$ Coefficient of Determination [17].
- *Step F:* A new set of weights is established by using a stochastic gradient, which is calculated by employing an optimizer function as follows: $\overline{W_{Updated}} = -\nabla * C(w) + \overline{W_{Initial}}$ [19].
- *Step G:* The revised weights are assigned to the various nodes and layers.
- *Step H:* The calculation process, as outlined from Steps B to G, is repeated until the algorithm learning rate no longer improves results or the output layer data reaches the acceptable tolerance range for each node [17,19,20].

## 3. ANN Modeling with Hyperparameters

While the ANN methodology is becoming popular in chemical engineering, there is limited information in the open technical literature, regarding the theoretically based selection of the best architectures for ANNs, in order to obtain VLL equilibrium thermodynamic data.

Given the above, Python version 3.9.5 was used to develop the ANN models of the present study. This coding software was selected given that it is an open-source tool, with a rich library of built-in modules for ML and AI, as well as with a good capability for result visualization. The main library modules used in this study include the following: (a) Scikit-learn (sklearn 0.0), (b) Keras 2.6.0, (c) Pandas 1.1.3, (d) NumPy 1.19.2, and (e) Matplotlib 3.3.2. Pandas and NumPy were used for data processing, Scikit-learn and Keras were applied for ML and AI tasks, and Matplotlib was employed to construct all the plots and graphs of this document.

Table 1 reports typical hyperparameters for the ANNs such as hidden layers, nodes, activation functions, optimizer, cost function, Epochs, and Batch sizes. These parameters are of importance as they provide the required accuracy for a wide range of possible computations. Thus, the development of an adequate ANN model must include the proper selection of the number of hidden layers, nodes, and activation functions [19]. On this basis, the determination of a successful ANN model is achieved with a good balance between accuracy and efficiency [12,18]. Table 1 also reports the advisable Epochs and Batch sizes, with Epoch size referring to the number of times that the algorithm repeatedly learns from the available training data [21], and Batch size referring to the number of mini-Epochs or Batches required for a given dataset.

**Table 1.** ANN Hyperparameters and their dataset types.

| Hyperparameter | Dataset Type | Usual Values |
|---|---|---|
| Hidden layer | Integer | 1, 2, 3 |
| Units (Neurons) | Integer | 10, 50, 100 |
| Activation Function | Equation | ReLu, Tanh, SoftMax |
| Optimizer | Algorithm | Adam, RMSP |
| Cost (or Loss) Function | Equation | MSE, MAE |
| Epoch | Integer | 50, 100, 200 |
| Batch Size | Integer | 2, 4, 8, 16, 32 |

Furthermore, in the present study, in order to establish the most appropriate ANN model, different hyperparameters were combined, creating at least six ANN models by employing different activation functions, as reported in Table 2.

**Table 2.** Activation functions—formulas and recommended applications.

| Activation Function | Equation | Recommended Applications [21] |
|---|---|---|
| ReLU | $\max(0, x)$ [22] | General purposes |
| ELU | $\begin{cases} x & , x \geq 0 \\ \alpha(e^x - 1) & , x < 0 \end{cases} ; 0 < \alpha < 1$ [21] | Classification |
| Sigmoid | $\frac{1}{1+e^{-x}}$ [12] | Binary classification |
| Tanh | $Tanh(x) = \frac{e^x - e^{-x}}{e^x + e^{-x}}$ [20] | Binary classification |
| SoftMax | $\frac{e^x}{\sum_{j=1}^{k} e^x}$ [12] | Multivariable classification |
| SoftPlus | $\log(e^x + 1)$ [22] | Function approximation |

The ELU, ReLU, Sigmoid, SoftMax, SoftPlus, and Tanh activation functions for the ANNs were evaluated by using several numbers of hidden layers, ranging from one to five and various numbers of nodes in every hidden layer, as follows: 5, 10, 50, 100, and 200. Regarding model configuration, the input layer involved two nodes and the output layer encompassed one node. This was the case for all the models considered, where the

temperature and hydrocarbon molar fraction were the input variables, and the pressure was the model output variable. Furthermore, model training was developed by using an 8 GB RAM memory, and a RYZEN 5-4000 series processor, with this taking 3.7 h of total computer time.

Table 3 describes various ANN hyperparameters adopted in the present study, used to develop ANN calculations.

**Table 3.** ANN hyperparameters.

| Hyperparameter | Value |
|---|---|
| Optimizer | Adam |
| Epochs | 100 |
| Batch Size | 32 |
| Cost (Loss) Function | MSE |

The Adam Optimizer hyperparameter from Table 3 was considered and evaluated by using Momentum and Root Mean Square Propagation, in order to establish the optimized descent gradients, which allow one to avoid the local minima and to calculate the global minimum [23,24] The ANN models evaluated included Epochs and Batch size hyperparameters. Epochs were set to 50, 100, and 150. It was shown that Epochs exceeding 100 did not improve the $R^2$ Coefficient of Determination significantly, as shown in Figure A2 in Appendix B. Thus, Epochs set to 100 were considered for all ANNs calculations. Furthermore, the Batch Size was set to 32. This is a typical batch size used to achieve training stability that requires a low computation time [25,26].

## 4. Evaluation of Proposed ANN Models

Data from the 61 runs developed in the CREC-VL-Cell led to 4200 records of cells, employed to evaluate the various ANN models considered, as described in Tables 2 and 3. Runs included conditions with input and output variable ranges, as described in Table 4.

**Table 4.** Description of Input and Output Variable Ranges involved in the ANN's Calculations of the present study.

| Data Inputs | Data Outputs |
|---|---|
| • Temperature: 70–110 °C <br> • Hydrocarbon molar Fractions: 0 to 1 | • Pressure: 100–400 KPa <br> • Phases: two or three <br> • Maximum Solubility for octane in water from 0.00015 to −0.00079 <br> • Maximum, solubility for water in octane from: 0.012 to 0.015 |

In this respect, the $R^2$ Coefficient of Determination helped to establish the ANN model's ability to provide adequate predictions. The $R^2$ was computed by means of the following equation:

$$\text{Coefficient of Determination} = R^2 = 1 - \frac{RSS}{TSS} = 1 - \frac{\sum_i (y_i - f_i)^2}{\sum_i (y_i - \overline{y})^2}, \quad (1)$$

where $RSS$ is the residual sum of squares, $TSS$ is the total sum of squares, $y_i$ is the output variable experimental data, $f_i$ is the output variable predicted data, and $\overline{y}$ is the predicted data mean [25]. Note that an $R^2$ Coefficient of Determination close to 1 is considered to be a good correlation, while a 0 value means that there is a very poor data correlation.

Figure 6 reports a comparative analysis of various ANN models evaluated by using various numbers of hidden layers and different activation functions. This graph shows how the number of hidden layers affects the $R^2$ Coefficient of Determination.

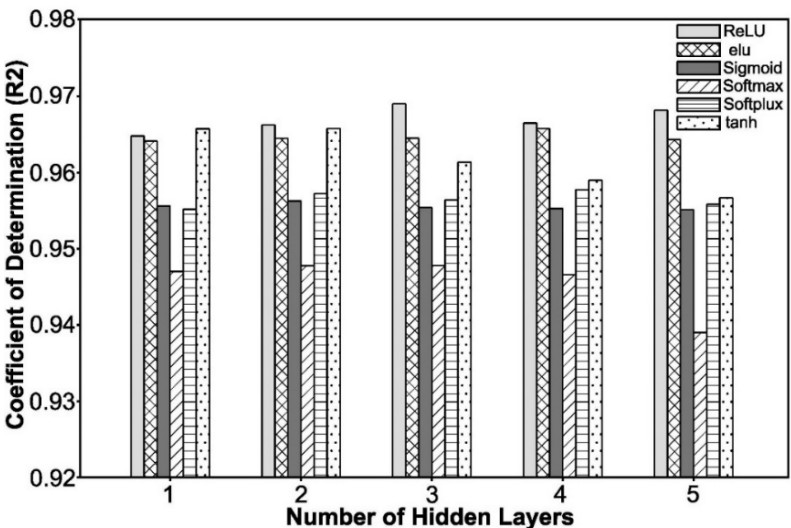

**Figure 6.** Change of the $R^2$ Coefficient of Determination ($R^2$) with the Number of Hidden Layers Using Different Activation Functions. Selected Hyperparameters: (a) Optimizer: Adam, (b) Epoch: 100, (c) Batch size: 32, (d) Number of nodes per hidden layer: 50.

It can be noted, as shown in Figure 6, that the ReLU activation function superseded all the others, in most cases, in term of the maximum value of the $R^2$ Coefficient of Determination, with this coefficient remaining within the 0.96–0.97 range, when the number of hidden layers increased from 1 to 5, with three hidden layers providing best results.

Furthermore, as shown in Figure 7, training computational times were significantly affected by the number of hidden layers. One interesting observation is that the ANNs with the ReLU activation function were, in most cases, the ones consistently requiring lower computational times.

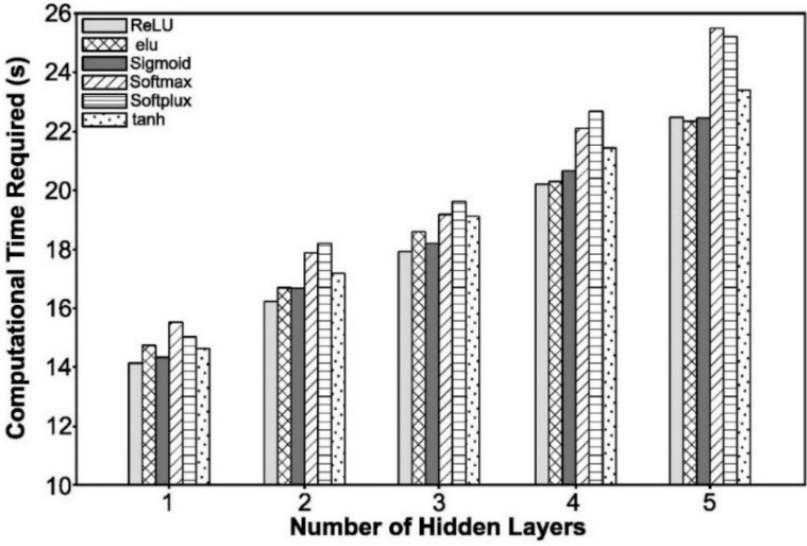

**Figure 7.** Average Computational Time(s) required for various numbers of hidden layers with different activation functions. Selected Hyperparameters: (a) Optimizer: Adam, (b) Epoch: 100, (c) Batch size: 32, and (d) Number of nodes per hidden layer: 50.

Figure 8 evaluates ANNs with various activation functions by varying the number of nodes per hidden layer. It shows how this affects the $R^2$ Coefficient of Determination. It was observed that the best ANNs were the ones using 50 nodes per hidden layer, with no significant improvement in the Coefficients of Determination when the number of nodes surpassed 50.

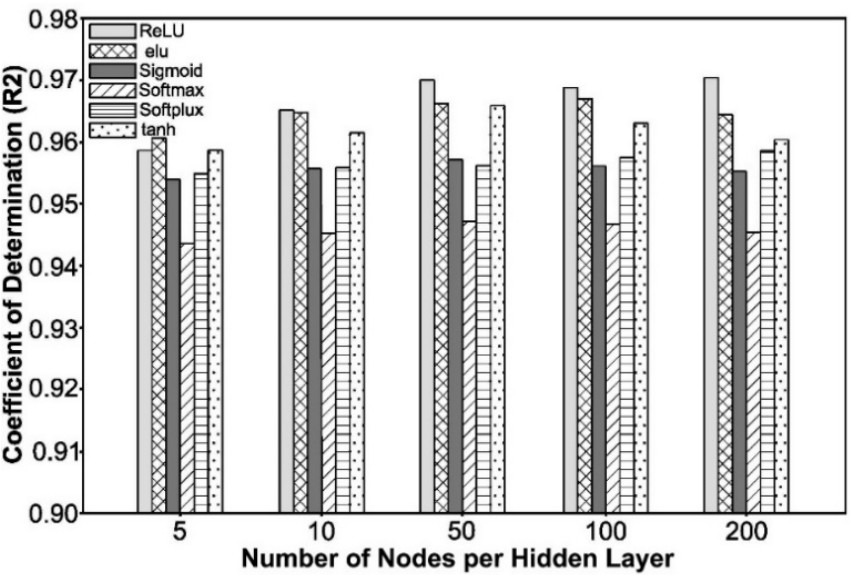

**Figure 8.** Change in the Coefficient of Determination ($R^2$) with various numbers of nodes per hidden layer, employing different activation functions. Selected Hyperparameters: (a) Optimizer: Adam, (b) Epoch: 100, (c) Batch size: 32, and (d) Number of hidden layers: 3.

In the case of Figure 8, it can be noted that the ANN that used the ReLU activation function displayed the best performance in terms of the $R^2$ Coefficient of Determination when compared to other ANN models using the other activation functions.

Finally, Figure 9 evaluates ANNs with various numbers of nodes per hidden layer, in terms of required computational training time.

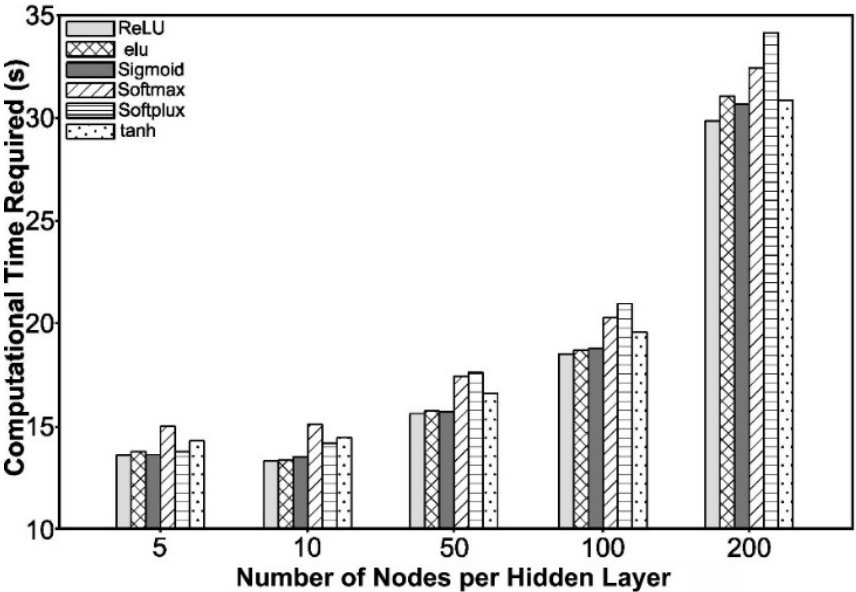

**Figure 9.** Average Computational Time(s) required for various numbers of nodes per hidden layer, using different activation functions. Selected Hyperparameters: (a) Optimizer: Adam, (b) Epoch:100, (c) Batch size: 32, and (d) Number of hidden layers: 3.

It can be observed, in Figure 9, that the ANN model with the ReLU activation function consistently required the shortest computational training time, when compared to ANNs with other activation functions.

Thus, on the basis of the above, it can be concluded that the best ANN, in the context of the present study, includes 3 hidden layers, 50 node units per hidden layer, and the ReLU

activation function. This conclusion was reached by observing the higher $R^2$ Coefficients of Determination obtained and the lower ANN computational training times required, when using these parameters. In this case, the $R^2$ Coefficient of Determination was very close to 0.97, and the required computational time was close to 15 s. It is interesting to note that the 0.97 $R^2$ Coefficient of Determination superseded the previous $R^2$ Coefficient of Determination values obtained via Machine Applications by our research team [6].

Given all this, an ANN designated as 3HL-50N-ReLU ANN (3 hidden layers, 50 nodes, and ReLU activation function) was selected to obtain the best system pressure predictions, as reported in the upcoming sections of this article.

## 5. Results and Discussions

The present study shows the viability of the ANNs used to predict the system pressure of highly diluted hydrocarbons in water. This is conducted without the need of defining a priori the number of liquid and vapor phases, present in complex hydrocarbon/water blends and found in naphtha recovery processes.

To train the ANN models, a system pressure dataset containing 4220 data points, resulting from various experiments was used. These records were from experiments conducted between 30 °C to 120 °C with octane in water concentrations ranging from 0 to 100 wt.% and with system pressures between 0 to 440 KPa.

To determine the accuracy of the ANN system pressure predictions at various temperatures and octane in water concentrations, the ANN was trained 20 times, while using a 70% randomly selected of the available experimental data. The remaining 30% of the experimental data was employed for validation purposes.

By using this approach, it was found that the average $R^2$ Coefficient of Determination for 30% of the experimental data was over 0.98 and that the average computational training time required for each model was 10.6 s. The lowest $R^2$ Coefficient of Determination was over 0.979 and the highest one was below 0.984, while the training times were in the range of 10.1 s to 11.5 s, respectively.

Figure 10 compares the predicted and experimentally measured system pressures when using the best-performing ReLU–ANN model, and 30% of the experimentally measured pressure values obtained with the CREC-VL-Cell. This figure displays an $R^2$ of 0.982 (average) with 11.1 s of required computational time.

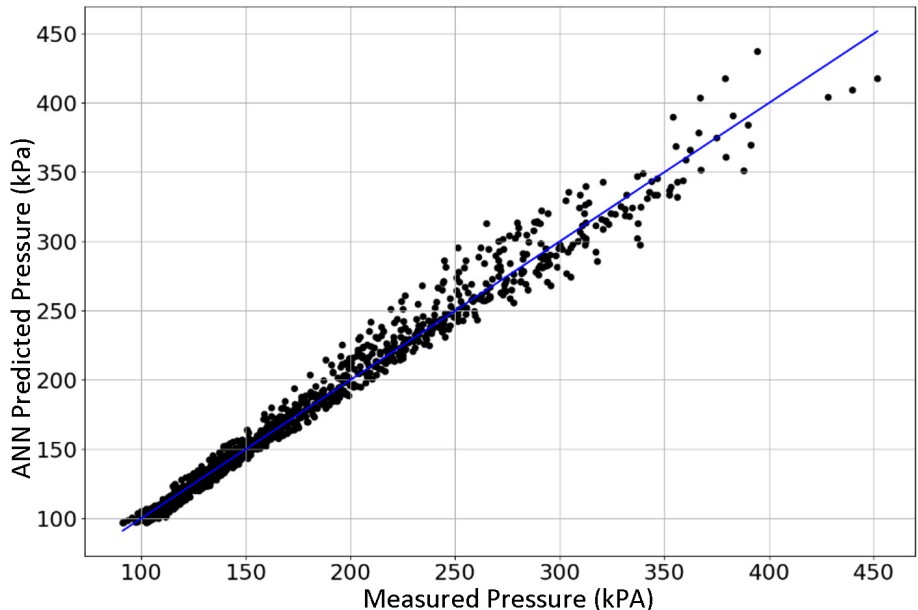

**Figure 10.** Comparison of predicted and experimentally Measured System Pressures, showing the accuracy of the proposed ReLU–ANN. Note: The experimental data reported corresponds to 30% of the data available, with the 70% remaining being radomly selected data and used for model training.

Thus, Figure 10 shows the excellent ability of the proposed ANN model to predict system pressure values with reasonable accuracy and limited computer time.

Once the proposed ANN model was trained and validated, 90% of the available experimental data was chosen randomly to create a restricted dataset. Then, the developed ANN model was retrained with 70% of the points of this restricted dataset. The remaining 30% of the limited-size dataset was employed for ANN validation. It was observed that under these conditions, the average $R^2$ Coefficient of Determination remained at 0.9439, with a minimum value of 0.9201 and a maximum value of 0.9583. Given, these good $R^2$ Coefficients of Determination obtained with 90% of the available data, it was considered that the original 4220 experimental data points were adequate for the ANN model development of the present study.

Figure 11a–d illustrates the adequacy of the proposed ANN model for predictions, with octane–water concentrations ranging from 0 to 100%, and temperatures ranging from 80 °C to 110 °C. Figure 11a–d report experimental data points [10], as well as pressure predictions obtained from the ANN model developed.

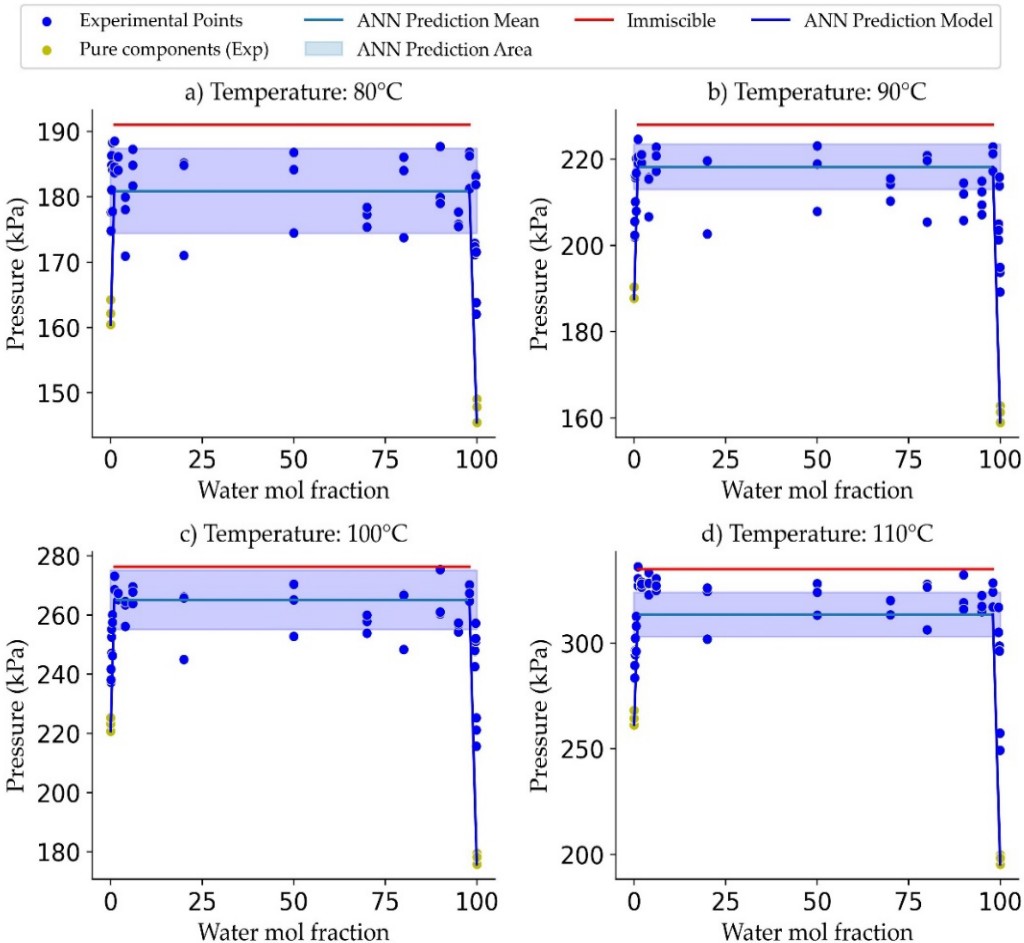

**Figure 11.** Changes of system pressure with temperature in the 80 °C to 110 °C range. Notes: (**a**) The red line describes the two-phase fully immiscible model, (**b**) the blue line shows the ANN average model predictions, (**c**,**d**) the blue band describes the data measured one standard deviations. Note: all $P_{mix}$ experimental and model derived points include the presence of air.

It can be observed that at all four thermal levels (80 °C, 90 °C, 100 °C, and 110 °C), the ANN model predicted the total pressure closely, for the eighteen octane in water concentrations ranging from 0 to 100%, with 7–9% standard deviations. One can thus conclude that the proposed ANN is adequate, without having to know in advance, the number of phases present, as required in our previous study [7].

Table 5 reports the maximum solubility of chemical species, for both water highly diluted in octane and in octane highly diluted in water, using the methodology proposed by Lopez Zamora (2021) [7]. Thus, the maximum solubility is obtained at the intersection point between the characteristic linear increasing pressure at very low and very high molar fractions, and the constant pressure, at all other intermediate molar fractions, as described in Figure 11a–d. It is speculated that at this intersection points, there is a change in the multiphase blend state, which evolves from two LV phases (liquid–vapour) to three LLV phases (liquid–liquid–vapour).

**Table 5.** Maximum Chemical Species Solubilities at Different Temperatures.

| Temperature | Maximum Water in Octane Solubilities | Maximum Octane in Water Solubilities |
|---|---|---|
| 80 °C | 0.012434 | 0.00015784 |
| 90 °C | 0.012871 | 0.00039511 |
| 100 °C | 0.014054 | 0.00039511 |
| 110 °C | 0.015615 | 0.00079189 |

One should note that results reported in Table 4 are in agreement with Lopez Zamora [7], who reported maximum solubilities for water highly diluted in octane and octane highly diluted in water, in the 0.014–0.017 and 0.00015–0.00079 ranges, respectively. These maximum solubilities are much higher (about 20 times) than the ones previously reported by others [27], adding significant value to the ANN calculations and data, reported in the present study.

## 6. Conclusions

- ANN models can be used effectively to calculate equilibrium system pressures in hydrocarbon–water blends.
- ANN models require the careful selection of ANN hyperparameters such as number of hidden layers, number of nodes per hidden layers, and activation functions.
- Three hidden layers, fifity node units per hidden layer, and the ReLU activation function are best parameters for the ANN model, yielding the highest $R^2$ Coefficients of Determination and requiring the shortest computational times.
- The implemented ReLU-ANN requires abundant VLL data for both training and validation. This required data was obtained experimentally using a CREC-VL-Cell, with octane–water concentrations, ranging from 0 to 100%, and temperatures ranging from 30 °C to 110 °C.
- The ANN–ReLU model used 70% randomly selected data from experiments in the CREC-VL-Cell for training, with the remaining 30% being available for ANN validation. This approach led to valuable ANN predictions, where an average $R^2$ Coefficient of Determination of 0.982 was obtained.
- The developed ANN-ReLu can be used to successfully predict both the system pressures in the entire range of octane–water compositions, as well as the maximum solubilities of octane highly diluted in water and water highly diluted in octane.

**Author Contributions:** Conceptualization, E.L.-R. and H.d.L.; methodology, E.L.-R., S.L.-Z., S.E. and H.d.L.; software studies, E.L.-R.; validation, E.L.-R., S.L.-Z. and S.E.; formal analysis, E.L.-R. and H.d.L.; investigation, E.L.-R.; re-sources, H.d.L.; data curation, S.E.; writing—original draft preparation, E.L.-R.; writing—review and editing, H.d.L., S.L.-Z. and S.E.; visualization, E.L.-R. and S.E.; supervision, H.d.L. and S.E.; project administration, H.d.L.; funding acquisition, H.d.L. All authors have read and agreed to the published version of the manuscript.

**Funding:** This research was funded by Natural Sciences and Engineering Research Council, Canada: HdL Discovery Grant; and Emerging Leaders America Program-Canada: E.Lopez Ramirez Scholarship.

**Acknowledgments:** The authors would like to acknowledge Jeonghoon Kong for the execution of the experiments in the CREC VL-Cell. The authors would also like to thank the Natural Science and Engineering Research Council of Canada (NSERC) and the Emerging Leaders in the Americas Program (ELAP), for the financial support provided for this work. Additionally, the authors would like to acknowledge Florencia de Lasa for her assistance with the editing of this manuscript.

**Conflicts of Interest:** The authors declare no conflict of interest.

## Nomenclature

Notation

| | |
|---|---|
| $a_{ij}$ | input variable in the "j" generic hidden layer |
| $C(w)$ | function involved in the stochastic gradient |
| $f_i$ | output variable predicted data. |
| $F(x)$ | Activation function |
| j | generic hidden layer |
| $m$ | Number of nodes |
| $P_{mix}$ | System pressure (KPa) |
| $RSS$ | Residual Sum of Squares |
| $R^2$ | Coefficient of Determination |
| $TSS$ | Total Sum of Squares |
| $x$ | Parameter is defined at each layer connection |
| $w_{ij}$ | Weights assigned in the generic "$j$" hidden layer |
| $\overline{W_{Initial}}$ | Stochastic gradient defined at the input of the hidden layer |
| $\overline{W_{Updated}}$ | Stochastic gradient defined at the input of the hidden layer |
| $y_i$ | Output variable experimental data |

*Greek Symbols*

| | |
|---|---|
| $\alpha(x)$ | Activation function parameter |

*Acronyms*

| | |
|---|---|
| AI | Artificial Intelligence |
| ANN | Artificial Neural Network |
| ML | Machine Learning |
| MSE | Mean Squared Error |
| MAE | Mean Absolute Error |
| VLE | Vapour-Liquid Equilibrium |
| VLLE | Vapour-Liquid-Liquid Equilibrium |

## Appendix A. Concentration and Temperature Level Data Required for ANN Model Training

*Appendix A.1. Concentration Effect*

Figure A1 reports a comparison of the ANN–ReLu model predictions and the experimental data obtained in the CREC-VL-Cell. The ANN model training was developed using a restricted experimental dataset, with two octane–water concentrations being randomly removed from the ANN calculations (0.001 and 0.8 water molar fractions). Again here, the ANN–ReLu was trained with a randomly selected 70% of the 3621 experimental data points, yielding a 0.982 average $R^2$ Coefficient of Determination, with minimum and maximum values of 0.993 and 0.97, respectively. One can thus conclude that even if two hydrocarbon–water concentrations from the available data, are removed, the accuracy of the ANN model is not affected, with $R^2$ Coefficients of Determination remaining in a very acceptable range.

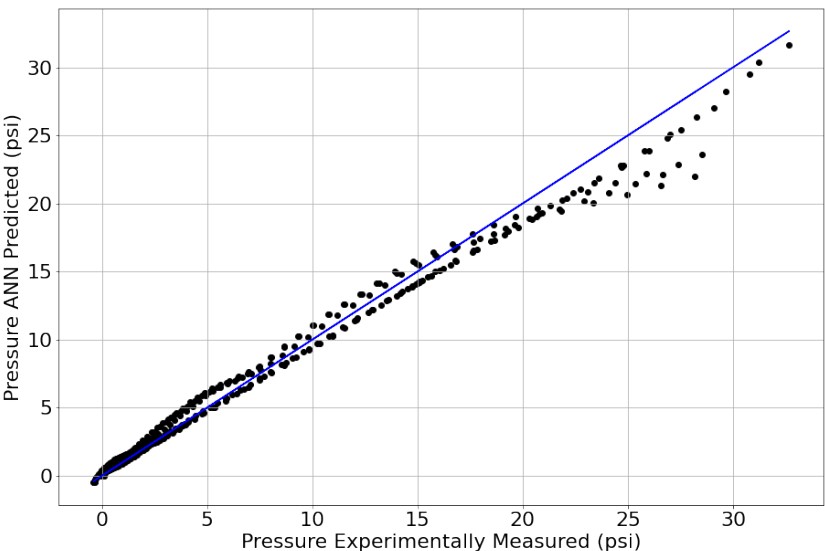

**Figure A1.** Comparison of the CREC-VL-Cell experimentally observed system pressures and the ANN–ReLu calculated pressures. Note: In the ANN model analysis two random octane–water concentrations were removed from the original dataset. Note: 414 records were used in this comparison.

*Appendix A.2. Temperature Effect*

Temperature is one of the two ANN input variables to predict the system pressure. In our study, the accuracy of the ANN–ReLu was evaluated by removing selectively data obtained at various temperature levels. For instance, when removing the data obtained in the 70 °C to 75 °C range and using an ANN–ReLu trained with 70% of all the experimental data records, it was observed that the average $R^2$ Coefficient of Determination was significantly reduced to an average of 0.3. This emphasizes the value of having a properly trained ANN that accounts for data, in the entire 30 °C to 120 °C range of interest.

## Appendix B. Pressure Mean Absolute Error Changes with the Number of Epochs

The proposed ANN model was evaluated using different number of the epochs or the equivalent number of iterations.

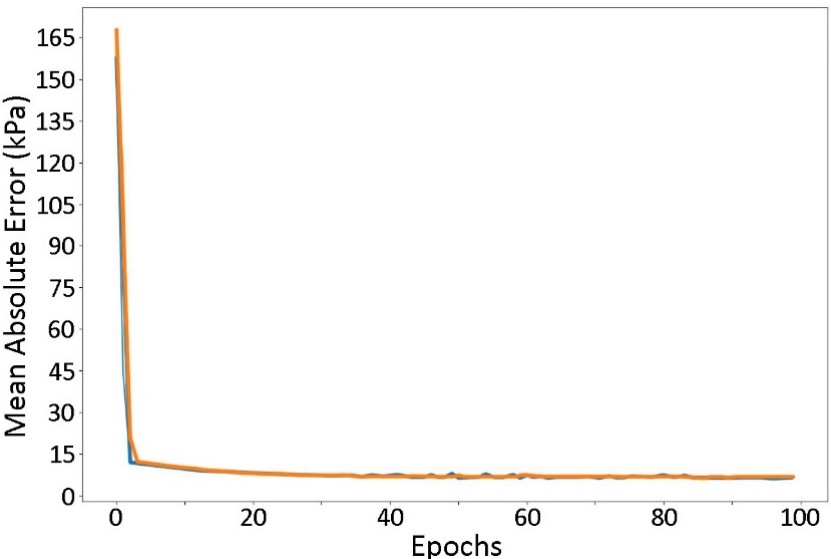

**Figure A2.** Changes of the Pressure Mean Absolute Error with the number of epochs. Notes: Blue curve represent the ANN results following training, and Orange curve describes the ANN results following model validation.

Figure A2 reports the ANN model mean absolute error during training as well as during model validation. It can be observed that mean absolute error decreases consistently with the number of epochs providing after 100 epochs a stable 6 KPa pressure mean absolute error. Thus, as result, to minimize the pressure mean absolute error 100 epochs were adopted in all calculation.

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
