# Peer review of "Artificial Neural Networks (ANNs) for Vapour-Liquid-Liquid Equilibrium (VLLE) Predictions in N-Octane/Water Blends"

_processes, doi:10.3390/pr11072026_

Round 1

Reviewer 1 Report

The authors have carried out investigation on: “Artificial Neural Networks (ANNs) for Vapour-Liquid-Liquid Equilibrium (VLLE) Predictions in N-Octane/Water Blends”, before acceptance the following Minor comments should be addressed.

Title: 1. Please rewrite the title in accordance with the theme of the paper.

Abstract: 1. Please avoid using abbreviations in the Abstract or use abbreviations with full form, mention it first time, then abbreviations can be used.

2. Please report the findings in the abstract, what is the enhancement?

Introduction: please check the grammar, typos throughout the introduction section, check the similarity index in the introduction section.

1. Please clearly add the Novelty statement at the end of the introduction section. Please add why the study is important and what are the outcomes of the study.

Material and Methodology:

1. Take care of subscripts and superscripts in Tables and abbreviations.

Results and discussion

1. Please improve the overall R and D section, please add previous studies to support your claims.

2. Please add from where the materials are procured.

3. Please add Artificial Intelligence (AI) VLLE calculations.

4. Add the statistical analysis.

5. Add the experimental data.

6. Please add the error analysis.

7. Add uncertainty in the results

Conclusion:

Please check the future scope and add relevance of the study, it will be better if you can add the conclusion in points for better understanding.

Author Response

Please refer to the enclosed file.

Regards.

Hugo de Lasa

Reviewer 2 Report

The paper generally acceptable. I have some comments in methodological context in general.

1) The introduction section requires slight modification. Please, list both research gaps and your contributions accordingly. Also, related works analysis needs further details while inserting a table summarizing obtained analysis results is required.

2) Please insert an introductory paragraph for sections before diving into subsections.

3) data description needs to provide further details by including data visualization for inputs and outputs after processing. This will provide insight of data complexity. Also, a table is necessary to describe data features.

4) A detailed description on hyperparameters tuning methodology is required. Meaning, what method did you used to select such hyperparameters and why?

5) is used to assess the performances of the model but it doesn’t give actually real assessment of differences between predicted and desired output. In this case I recommend first adding a detailed description about using this metric. Second, please consider further analysis criteria and further metrics.

6) It is recommended to compare proposed solution to state of the art methods.

Author Response

(The authors gave the same response as above.)

Reviewer 3 Report

1. The abstract should be supported by some results from the main part of the paper. 

2. The innovation extracted from the paper is not clear enough.

3. Not all abbreviations are explained. 

4. Check the grammar (row 81 instead of or is ir), etc.

5. There are sections where the text is aligned into the right margin, it is supposed to be evenly between the margins.

6. In my view instead of describing general ANN more attention should be paid in to the particular ANN application for VLLE predictions (maybe the Figure with ANN architecture). 

7. References are not as per the guidelines of the journal. 

The grammar should be check. 

Author Response

(The authors gave the same response as above.)

Round 2

Reviewer 3 Report

The manuscript is revised according the comments, however the references are still not as per the guidelines of the journal.